# DISK: Learning local features with policy gradient

**Michał J. Tyszkiewicz**[1]     **Pascal Fua**[1]     **Eduard Trulls**[2]
[1]École Polytechnique Fédérale de Lausanne (EPFL)     [2]Google Research, Zurich
michal.tyszkiewicz@epfl.ch    pascal.fua@epfl.ch    trulls@google.com

## Abstract

Local feature frameworks are difficult to learn in an end-to-end fashion, due to the *discreteness* inherent to the selection and matching of sparse keypoints. We introduce DISK (DIScrete Keypoints), a novel method that overcomes these obstacles by leveraging principles from Reinforcement Learning (RL), optimizing end-to-end for a high number of correct feature matches. Our simple yet expressive probabilistic model lets us keep the training and inference regimes close, while maintaining good enough convergence properties to reliably train from scratch. Our features can be extracted very densely while remaining discriminative, challenging commonly held assumptions about what constitutes a good keypoint, as showcased in Fig. 1, and deliver state-of-the-art results on three public benchmarks.

## 1  Introduction

Local features have been a key computer vision technology since the introduction of SIFT [20], enabling applications such as Structure-from-Motion (SfM) [1, 15, 36], SLAM [27], re-localization [23], and many others. While not immune to the deep learning "revolution", 3D reconstruction is one of the last bastions where sparse, hand-crafted solutions remain competitive with or outperform their dense, learned counterparts [37, 34, 16]. This is due to the difficulty of designing end-to-end methods with a differentiable training objective that corresponds well enough with the downstream task.

While patch descriptors can be easily learned on predefined keypoints [38, 39, 25, 40, 13], joint detection and matching is harder to relax in a differentiable manner, due to its computational complexity. Given two images $A$ and $B$ with feature sets $F_A$ and $F_B$, matching them is $O(|F_A| \cdot |F_B|)$. As each image pixel may become a feature, the problem quickly becomes intractable. Moreover, the "quality" of a given feature depends on the rest, because a feature that is very similar to others is less distinctive, and therefore less useful. This is hard to account for during training.

We address this issue by bridging the gap between training and inference to fully leverage the expressive power of CNNs. Our backbone is a network that takes images as input and outputs keypoint 'heatmaps' and dense descriptors. Discrete keypoints are sampled from the heatmap, and the descriptors at those locations are used to build a distribution over feature matches across images. We then use geometric ground truth to assign positive or negative rewards to each match, and perform gradient descent to maximize the expected reward $\mathbb{E} \sum_{(i,j) \in M_{A \leftrightarrow B}} r(i \leftrightarrow j)$, where $M_{A \leftrightarrow B}$ is the set of matches and $r$ is per-match reward. In effect, this is a policy gradient method [44].

Probabilistic relaxation is powerful for discrete tasks, but its applicability is limited by the fact that the expected reward and its gradients usually cannot be computed exactly. Therefore, noisy Monte Carlo approximations have to be used instead, which harms convergence. We overcome this difficulty by careful modeling that yields analytical expressions for the gradients. As a result, we can benefit from the expressiveness of policy gradient, narrowing the gap between training and inference and ultimately outperforming state-of-the-art methods, while still being able to train models from scratch.

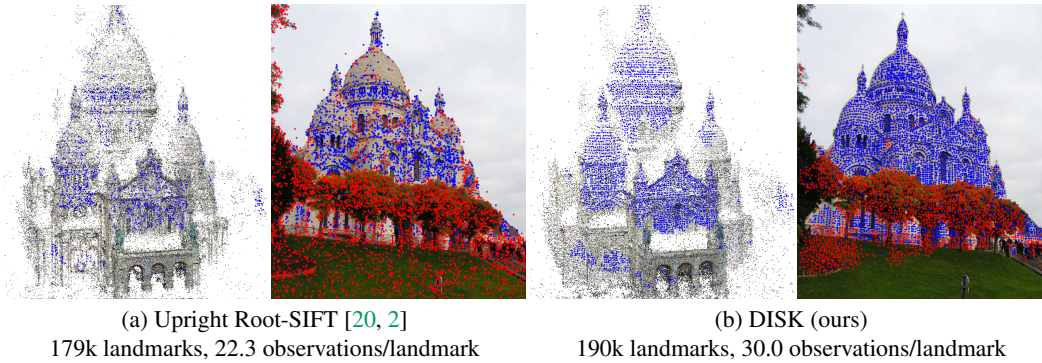

(a) Upright Root-SIFT [20, 2]
179k landmarks, 22.3 observations/landmark

(b) DISK (ours)
190k landmarks, 30.0 observations/landmark

Figure 1: **SIFT vs. DISK in SfM.** We reconstruct "Sacre Coeur" from 1179 images [16] with COLMAP. For Upright Root-SIFT (left) and DISK (right) we show a point cloud and one image with its keypoints. Landmarks, and their respective keypoints, are drawn in **blue**. Keypoints which do not create landmarks are drawn in **red**. Our features can be extracted (and create associations) on seemingly textureless regions where SIFT fails to, producing more landmarks with more observations.

Our contribution therefore is a novel, end-to-end-trainable approach to learning local features that relies on policy gradient. It yields considerably more accurate matches than earlier methods, and this results in better performance on downstream tasks, as illustrated in Fig. 1 and Sec. 4.

## 2   Related Work

The process of extracting local features usually involves three steps: finding a keypoint, estimating its orientation, and computing a description vector. In traditional methods such as SIFT [20] or SURF [4], this involves many hand-crafted heuristics. The first wave of local features involving deep networks featured descriptors learned from patches extracted on SIFT keypoints [48, 14, 38] and some of their successors, such as HardNet [25], SOSNet [40], and LogPolarDesc [13], are still state-of-the-art. Other learning-based methods focus on keypoints [42, 35, 18] or orientations [47], or merge the two notions entirely [8].

These methods attack a single element of this process. Others have developed end-to-end-trainable pipelines [45, 10, 29, 11, 31] that can optimize the whole process and, hopefully, improve performance. However, they either use inexact approximations to the true objective [10, 31], break differentiability [29] or make big assumptions, such as extrema in descriptor space making good features [11].

Three recent approaches are attempting to bridge the gap between training and inference in a spirit close to ours. GLAMpoints [41] seeks to estimate homographies between retinal images and use Reinforcement Learning (RL) methods to find keypoints that are correctly matched by SIFT descriptors. Since matching is deterministic, Q-learning can be used to regress for the expected reward of each keypoint, rather than optimize directly in policy space. Using hand-crafted descriptors and only addressing the detection problem was motivated by domain-specific requirements of strong rotation equivariance, which most learned models lack. While it makes sense in the specific scenario it was developed for, it limits what the method can do. Similarly, [9] also uses handcrafted descriptors and learns to predict the probability that each pixel would be successfully matched with those. Their approach therefore inherits many of the limitations of GLAMpoints.

Reinforced Feature Points [6] address the more difficult issue of learning with a general non-differentiable objective for the purpose of camera pose estimation, with RANSAC in the loop. Unfortunately, supervising all detection and matching decisions with a single reward means that this approach suffers from weak training signal, an endemic RL problem, and has to rely on pre-trained models from [10] that can only be fine-tuned. Our method can be seen as a relaxation of their approach, where we train for a surrogate objective: finding many correct feature matches. This allows for substantially more robust training from scratch and yields better downstream results.

## 3   Method

Given images $A$ and $B$, our goal is first to extract a set of local features $F_A$ and $F_B$ from each and then match them to produce a set of correspondences $M_{A\leftrightarrow B}$. To learn how to do this through reinforcement learning, we redefine these two steps probabilistically. Let $P(F_I|I, \theta_F)$ be a distribution over sets of features $F_I$, conditional on image $I$ and feature detection parameters $\theta_F$, and $P(M_{A\leftrightarrow B}|F_A, F_B, \theta_M)$ be a distribution over matches between features in images $A$ and $B$, conditional on features $F_A$, $F_B$, and matching parameters $\theta_M$. Calculating $P(M_{A\leftrightarrow B}|A, B, \theta)$ and its derivatives requires integrating the product of these two probabilities over all possible $F_A$, $F_B$, which is clearly intractable. However, we can estimate gradients of expected reward $\nabla_\theta \mathbb{E}_{M_{A\leftrightarrow B} \sim P(M_{A\leftrightarrow B}|A,B,\theta)} R(M_{A\leftrightarrow B})$ via Monte Carlo sampling and use gradient ascent to maximize that quantity.

**Feature distribution** $P(F_I|I, \theta_F)$**.** Our feature extraction network is based on a U-Net [32], with one output channel for detection and $N$ for description. We denote these feature maps as $\mathbf{K}$ and $\mathbf{D}$, respectively, from which we extract features $F = \{K, D\}$. We pick $N$=128, for a direct comparison with SIFT and nearly all modern descriptors [20, 25, 21, 40, 13, 31].

The detection map $\mathbf{K}$ is subdivided into a grid with cell size $h \times h$, and we select at most one feature per grid cell, similarly to SuperPoint [10]. To do so, we crop the feature map corresponding to cell $u$, denoted $\mathbf{K}^u$, and use a softmax operator to normalize it. Our probabilistic framework samples a pixel $p$ in cell $u$ with probability $P_s(p|\mathbf{K}^u) = \text{softmax}(\mathbf{K}^u)_p$. This detection proposal $p$ may still be rejected: we accept it with probability $P_a(\text{accept}_p|\mathbf{K}^u) = \sigma(\mathbf{K}^u_p)$, where $\mathbf{K}^u_p$ is the (scalar) value of the detection map $\mathbf{K}$ at location $p$ in cell $u$, and $\sigma$ is a sigmoid. Note that $P_s(p|\mathbf{K}^u)$ models *relative* preference across a set of different locations, whereas $P_a(\text{accept}_p|\mathbf{K}^u)$ models the *absolute* quality for location $p$. The total probability of sampling a feature at pixel $p$ is thus $P(p|\mathbf{K}^u) = \text{softmax}(\mathbf{K}^u)_p \cdot \sigma(\mathbf{K}^u_p)$. Once feature locations $\{p_1, p_2, ...\}$ are known, we associate them with the $l_2$-normalized descriptors at this location, yielding a set of features $F_I = \{(p_1, \mathbf{D}(p_1)), (p_2, \mathbf{D}(p_2)), ...\}$. At inference time we replace softmax with $\arg\max$, and $\sigma$ with the sign function. This is again similar to [10], except that we retain the spatial structure and interpret cell $\mathbf{K}^u$ in both a relative and an absolute manner, instead of creating an extra *reject* bin.

**Match distribution** $P(M_{A\leftrightarrow B}|F_A, F_B, \theta_M)$**.** Once feature sets $F_A$ and $F_B$ are known, we compute the $l_2$ distance between their descriptors to obtain a distance matrix $\mathbf{d}$, from which we can generate matches. In order to learn good local features it is crucial to refrain from matching ambiguous points due to repeated patterns in the image. Two solutions to this problem are cycle-consistent matching and the ratio test. Cycle-consistent matching enforces that two features be nearest neighbours of each other in descriptor space, cutting down on the number of putative matches while increasing the ratio of correct ones. The ratio test, introduced by SIFT [20], rejects a match if the ratio of the distances between its first and second nearest neighbours is above a threshold, in order to only return confident matches. These two approaches are often used in conjunction and have been shown to drastically improve results in matching pipelines [5, 16], but they are not easily differentiable.

Our solution is to relax cycle-consistent matching. Conceptually, we draw *forward* (A→B) matches for features $F_{A,i}$ from categorical distributions defined by the rows of distance matrix $\mathbf{d}$, and *reverse* (A←B) matches for features $F_{B,j}$ from distributions based on its columns. We declare $F_{A,i}$ to match $F_{B,j}$ if both the forward and reverse matches are sampled, *i.e.*, if the samples are consistent. The forward distribution of matches is given by $P_{A\rightarrow B}(j|\mathbf{d}, i) = \text{softmax}\left(-\theta_M \mathbf{d}(i, \cdot)\right)_j$, where $\theta_M$ is the single parameter, the inverse of the softmax temperature. $P_{A\leftarrow B}$ is analogously defined by $\mathbf{d}^T$.

It should be noted that, given features $F_A$ and $F_B$, the probability of any particular match can be computed *exactly*: $P(i \leftrightarrow j) = P_{A\rightarrow B}(i|\mathbf{d}, j) \cdot P_{A\leftarrow B}(j|\mathbf{d}, i)$. Therefore, as long as reward $R$ factorizes over matches as $R(M_{A\leftrightarrow B}) = \sum_{(i,j)\in M_{A\leftrightarrow B}} r(i \leftrightarrow j)$, given $F_A$ and $F_B$, we can compute *exact* gradients $\nabla_{D,\theta_M} \mathbb{E} R(M_{A\leftrightarrow B})$, without resorting to sampling. This means that the matching step does not contribute to the overall variance of gradient estimation, unlike in [6], which we believe to be key to the good convergence properties of our model. Finally, one can also replace our matching relaxation with a non-probabilistic loss like in [25]. While it may be superior for descriptors alone, our solution upholds the probabilistic interpretation of the pipeline, making the hyperparameters $(\lambda_{tp}, \lambda_{fp}, \lambda_{kp})$ easy to tune and naturally integrating with the gradient estimation in keypoint detection.

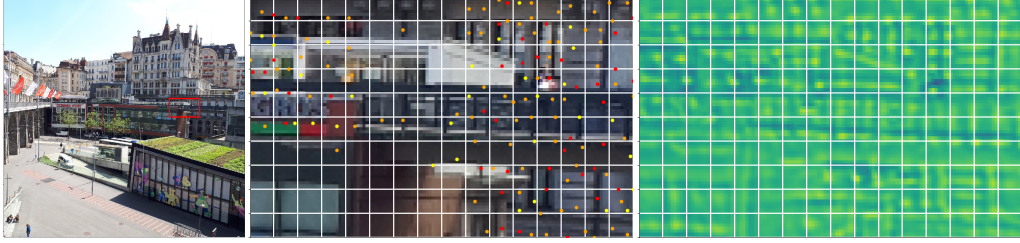

Figure 2: **Non-Maxima Suppression vs Grid-based sampling.** We demonstrate the benefits of replacing the 1-per-cell sampling approach used during training with simple NMS at inference time. For a small region of an image (left), marked by the red box, we show the features chosen through NMS (middle) and the 'heatmap' $\mathbf{K}$ (right), overlaid by the grid. Notice how maxima can be cut by cell boundaries. Keypoints are sorted by "score" and color-coded: the top third are drawn in **red**, the next third in **orange**, and the rest in **yellow**. Each cell contains at most two very salient (red) features.

**Reward function $R(M_{A\leftrightarrow B})$.** As stated above, if the reward $R(M_{A\leftrightarrow B})$ can be factorized as a sum over individual matches, the formulation of $P(M_{A\leftrightarrow B}|F_A, F_B, \theta_M)$ allows for the use of closed-form formulas while training. For this reason we use a very simple reward, which rewards correct matches with $\lambda_{\text{tp}}$ points and penalizes incorrect matches with $\lambda_{\text{fp}}$ points. Let's assume we have ground-truth poses and pixel-to-pixel correspondences in the form of depth maps. We declare a match *correct* if depth is available at both $p_{A,i}$ and $p_{B,j}$, and both points lie within $\epsilon$ pixels of their respective reprojections. We declare a match *plausible* if depth is not available at either location, but the epipolar distance between the points is less than $\epsilon$ pixels, in which case we neither reward nor penalize it. We declare a match *incorrect* in all other cases.

**Gradient estimator.** With $R$ factorized over matches and $P(i \leftrightarrow j|F_A, F_B, \theta_M)$ given as a closed formula, the application of the basic policy gradient [44] is fairly simple: with $F_A, F_B$ sampled from their respective distributions $P(F_A|A, \theta_F), P(F_B|B, \theta_F)$ we have

$$\nabla_\theta \mathop{\mathbb{E}}_{M_{A\leftrightarrow B}} R(M_{A\leftrightarrow B}) = \mathop{\mathbb{E}}_{F_A, F_B} \sum_{i,j} [P(i \leftrightarrow j|F_A, F_B, \theta_M) \cdot r(i \leftrightarrow j) \cdot \nabla_\theta \Gamma_{ij}], \quad (1)$$

$$\text{where } \Gamma_{ij} = \log P(i \leftrightarrow j|F_A, F_B, \theta_M) + \log P(F_{A,i}|A, \theta_F) + \log P(F_{B,j}|B, \theta_F).$$

The summation above is non-exhaustive, missing the case of $i$ not being matched with any $j$: since we award non-matches 0 reward, they can be safely ommited from the gradient estimator. Having a closed formula for $P(i \leftrightarrow j|F_A, F_B, \theta_M)$ along with $R$ being a sum over individual matches allows us to compute the sum in equation 1 exactly, which in the general case of REINFORCE [44] would have to be replaced with an empirical expectation over sampled matches, introducing variance in the gradient estimates. In our formulation, the only sources of gradient variance are due to mini-batch effects and approximating the expectation w.r.t. choices of $F_A, F_B$ with an empirical sum.

It should also be noted that our formulation does not provide the feature extraction network with any supervision other than through the quality of matches those features participate in, which means that a keypoint which is never matched is considered neutral in terms of its value. This is a very useful property because keypoints may not be co-visible across two images, and should not be penalized for it as long as they do not create incorrect associations. On the other hand, this may lead to many unmatchable features on clouds and similar non-salient structures, which are unlikely to contribute to the downstream task but increase the complexity in feature matching. We address this by imposing an additional, small penalty on each sampled keypoint $\lambda_{\text{kp}}$, which can be thought of as a regularizer.

**Inference.** Once the models have been trained we discard our probabilistic matching framework in favor of a standard cycle-consistency check, and apply the ratio test with a threshold found empirically on a validation set. Another consideration is that our method is confined to a grid, illustrated in Fig. 2. This has two drawbacks. Firstly, it can sample at most one feature per cell. Secondly, each cell is blind to its neighbours. Our method may thus select two contiguous pixels as distinct keypoints. At inference time we can work around this issue by applying non-maxima suppression on the feature map $\mathbf{K}$, returning features at all local maxima. This addresses both issues at the cost of a misalignment between training and inference, which is potentially sub-optimal. We discuss this further in Sec. 4.4.

# 4 Experiments

We first describe our specific implementation and the training data we rely on. We then evaluate our approach on three different benchmarks, and present two ablation studies.

**Training data.** We use a subset of the MegaDepth dataset [19], from which we choose 135 scenes with 63k images in total. They are posed with COLMAP, a state-of-the-art SfM framework that also provides dense depth estimates we use to establish pixel-to-pixel correspondences. We omit scenes that overlap with the test data of the Image Matching Challenge (Sec. 4.1), and apply a simple co-visibility heuristic to sample viable pairs of images. See the supplementary material for details.

**Feature extraction network.** We use a variation of the U-Net [32] architecture. Our model has 4 down- and up-blocks which consist of a single convolutional layer with $5 \times 5$ kernels, unlike the standard U-Net that uses two convolutional layers per block. We use instance normalization instead of batch normalization, and PReLU non-linearities. Our models comprise 1.1M parameters, with a formal receptive field of $219 \times 219$ pixels. Training and inference code is available at https://github.com/cvlab-epfl/disk.

**Optimization.** Although the matching stage has a single learnable parameter, $\theta_M$, we found that gradually increasing it with a fixed schedule works well, leaving just the feature extraction network to be learned with gradient descent. Since the training signal comes from *matching features*, we process three co-visible images A, B and C per batch. We then evaluate the summation part of equation 1 for pairs $A \leftrightarrow B$, $A \leftrightarrow C$, $B \leftrightarrow C$ and accumulate the gradients w.r.t. $\theta$. While matching is pair-wise, we obtain three image pairs per image triplet. By contrast, two pairs of unrelated scenes would require four images. Our approach provides more matches while reducing GPU memory for feature extraction. We rescale the images such that the longer edge has 768 pixels, and zero-pad the shorter edge to obtain a square input; otherwise we employ no data augmentation in our pipeline. Grid cells are square, with each side $h = 8$ pixels.

Rewards are $\lambda_{\text{tp}} = 1$, $\lambda_{\text{fp}} = -0.25$ and $\lambda_{\text{kp}} = -0.001$. Since a randomly initialized network tends to generate very poor matches, the quality of keypoints is negative on average at first, and the network would cease to sample them at all, reaching a local maximum reward of 0. To avoid that, we anneal $\lambda_{\text{fp}}$ and $\lambda_{\text{kp}}$ over the first 5 epochs, starting with 0 and linearly increasing to their full value at the end.

We use a batch of two scenes, with three images in each. Since our model uses instance normalization instead of batch normalization, it is also possible to accumulate gradients over multiple smaller batches, if GPU memory is a bottleneck. We use ADAM [17] with learning rate of $10^{-4}$. To pick the best checkpoint, we evaluate performance in terms of pose estimation accuracy in stereo, with DEGENSAC [7]. Specifically, every 5k optimization steps we compute the mean Average Accuracy (mAA) at a $10^o$ error threshold, as in [16]: see Sec. 4.1 and the appendix for details.

Finally, our method produces a variable number of features. To compare it to others under a fixed feature budget, we subsample them by their "score", that is, the value of heatmap $\mathbf{K}$ at that location.

## 4.1 Evaluation on the 2020 Image Matching Challenge (IMC) [16] – Table 1, Figures 3 and 4

The Image Matching Challenge provides a benchmark that can be used to evaluate local features for two tasks: stereo and multi-view reconstruction. For the stereo task, features are extracted across every pair of images and then given to RANSAC, which is used to compute their relative pose. The multiview task uses COLMAP to generate SfM reconstructions from small subsets of 5, 10, and 25 images. The differentiating factor for this benchmark is that both tasks are evaluated *downstream*, in terms of the quality of the reconstructed poses, which are compared to the ground truth, by using the mean Average Accuracy (mAA) up to a 10-degree error threshold. While this requires carefully tuning components extraneous to local features, such as RANSAC hyperparameters, it measures performance on real problems, rather than intermediate metrics.

**Hyperparameter selection.** We rely on a validation set of two scenes: "Sacre Coeur" and "St. Peter's Square". We resize the images to 1024 pixels on the longest edge, generate cycle-consistent matches with the ratio test, with a threshold of 0.95. For stereo we use DEGENSAC [7], which outperforms vanilla RANSAC [16], with an inlier threshold of 0.75 pixels.

| Method | Up to 2048 features/image | | | | | | | Up to 8000 features/image | | | | | | |
| | Task 1: stereo | | | Task 2: Multiview | | | | Task 1: stereo | | | Task 2: Multiview | | | |
| | NM | NI | mAA(10°) | NM | NL | TL | mAA(10°) | NM | NI | mAA(10°) | NM | NL | TL | mAA(10°) |
|---|---|---|---|---|---|---|---|---|---|---|---|---|---|---|
| Upright Root-SIFT | 194.0 | 112.3 | 0.3986 | 199.3 | 1341.7 | 4.09 | 0.5623 | 525.4 | 358.9 | 0.5075 | 542.9 | 4404.6 | 4.38 | 0.6792 |
| Upright L2-Net | 174.1 | 117.1 | 0.4192 | 179.8 | 1361.3 | 4.23 | 0.5968 | 657.3 | 435.7 | 0.5450 | 395.5 | 3603.8 | 4.38 | 0.6849 |
| Upright HardNet | 274.0 | 152.7 | **0.4609** | 201.3 | 1467.9 | 4.31 | 0.6354 | 791.7 | 527.6 | 0.5728 | 509.1 | 4250.4 | 4.55 | 0.7231 |
| Upright GeoDesc | 235.8 | 132.7 | 0.4136 | 161.1 | 1287.3 | 4.24 | 0.5837 | 598.9 | 409.9 | 0.5267 | 458.6 | 4146.8 | 4.41 | 0.7044 |
| Upright SOSNet | 265.6 | 171.2 | 0.4505 | 194.0 | 1442.3 | 4.31 | 0.6359 | 752.9 | 508.4 | 0.5738 | 464.4 | 3988.6 | 4.52 | 0.7129 |
| Upright LogPolarDesc | 296.8 | 162.2 | 0.4567 | 211.9 | 1553.4 | 4.33 | 0.6370 | 821.7 | 543.2 | 0.5510 | 505.4 | 4414.1 | 4.52 | 0.7109 |
| SuperPoint | 292.8 | 126.8 | 0.2964 | 169.3 | 1184.3 | 4.34 | 0.5464 | – | – | – | – | – | – | – |
| LF-Net | 191.1 | 106.5 | 0.2344 | 196.7 | 1385.0 | 4.14 | 0.5141 | – | – | – | – | – | – | – |
| D2-Net (SS) | 505.7 | 188.4 | 0.1813 | 513.1 | **2357.9** | 3.39 | 0.3943 | 1258.2 | 482.3 | 0.2228 | 1278.7 | 5893.8 | 3.62 | 0.4598 |
| D2-Net (MS) | 327.8 | 134.8 | 0.1355 | 337.6 | **2177.3** | 3.01 | 0.3007 | 1028.6 | 470.6 | 0.2506 | 1054.7 | **6759.3** | 3.39 | 0.4751 |
| R2D2 | 273.6 | 213.9 | 0.3346 | 280.8 | 1228.4 | 4.29 | 0.6149 | 1408.8 | **842.2** | 0.4437 | 739.8 | 4432.9 | 4.59 | 0.6832 |
| Submission #609 | 439.7 | **270.0** | **0.4690** | 280.4 | 1489.6 | **4.69** | **0.6812** | – | – | – | – | – | – | – |
| Submission #578 | 439.5 | **246.6** | 0.4542 | 331.6 | 1621.7 | **4.57** | **0.6741** | – | – | – | – | – | – | – |
| Submission #599 | 227.4 | 129.5 | 0.4507 | 176.6 | 1209.6 | 4.44 | 0.6609 | – | – | – | – | – | – | – |
| Submission #611 | – | – | – | – | – | – | – | 945.4 | 622.1 | **0.5887** | 899.1 | 6086.2 | **4.65** | **0.7513** |
| Submission #613 | – | – | – | – | – | – | – | 934.9 | **624.1** | 0.5873 | 964.8 | **6350.7** | 4.64 | **0.7495** |
| Submission #625 | – | – | – | – | – | – | – | 945.4 | 605.1 | **0.5878** | 899.1 | 6095.8 | **4.65** | 0.7485 |
| DISK (#708 & #709) | 514.2 | **404.2** | **0.5132** | 527.5 | **2428.0** | **5.55** | **0.7271** | 1621.9 | **1238.5** | 0.5585 | 1663.8 | **7484.0** | **5.92** | **0.7502** |
| Δ (%) | +1.7 | +49.7 | +9.4 | +2.8 | +3.0 | +18.3 | +6.7 | +15.1 | +47.1 | -5.4 | +30.1 | +10.7 | +27.3 | -0.1 |

Table 1: **Image Matching Challenge results.** The primary metric is **(mAA)**, the mean Average Accuracy in pose estimation, up to $10^o$. We also report **(NM)** the number of matches (given to RANSAC for stereo, and to COLMAP for multiview). For stereo, we also report **(NI)** the number of RANSAC inliers. For multiview, we also report **(NL)** number of landmarks (3D points), and **(TL)** track length (observations per landmark). The top 3 results are highlighted in **red**, **green** and **blue**.

**Results.** We extract DISK features for the nine test scenes, for which the ground truth is kept private, and submit them to the organizers for processing. The challenge has two categories: up to 2k or 8k features per image. We participate in both. We report the results in Table 1, along with baselines taken directly from the leaderboards, computed in [16]. We consider several descriptors on DoG keypoints: RootSIFT [20, 2] L2-Net [39], HardNet [25], GeoDesc [22], SOSNet [40] and LogPolarDesc [13]. For brevity, we show only their upright variants, which perform better than their rotation-sentitive counterparts on this dataset. For end-to-end methods, we consider SuperPoint [10], LF-Net [29], D2-Net [11], and R2D2 [31]. All of these methods use DEGENSAC [7] as a RANSAC variant for stereo, with their optimal hyperparameters. We also list the top 3 user submissions for each category, taken from the leaderboards on June 5, 2020 (the challenge concluded on May 31, 2020).

On the 2k category, we outperform all methods by 9.4% relative in stereo, and 6.7% relative in multiview. On the 8k category, averaging stereo and multiview, we outperform all baselines, but place slightly below the top three submissions. Our method can find many more matches than any other, easily producing 2-3x the number of RANSAC inliers or 3D landmarks. Our features used for the 2k category are a subset of those used for 8k, which indicates a potentially sub-optimal use of the increased budget, which may be solved training with larger images or smaller grid cells. We show qualitative images in Figs. 3 and 4. Further results are available in the supplementary material.

Note that we only compare with submissions using the built-in feature matcher, based on the $l_2$ distance between descriptors, instead of neural-network based matchers [46, 49, 33], which combined with state-of-the-art features obtain the best overall results. Even so, DISK places #2 below only SuperGlue [33] on the 2k category, outperforming *all other solutions* using learned matchers.

**Rotation invariance.** We observe our models break under large in-plane rotations, which is to be expected. We evaluate their performance with an additional test using synthetic data. We pick 36 images randomly from the IMC 2020 validation set, match them with their copies, rotated by $\theta$, and calculate the ratio of correct matches, defined as those below a 3-pixel reprojection threshold. In Fig. 6 we report it for different state-of-the-art methods that, like ours, bypass orientation detection, and overlay a histogram of the differences in in-plane rotation in the dataset. We find that DISK is exceptionally robust to the range of rotations it was exposed to, and loses performance outside of this range, suggesting that failure modes such as in Fig. 3 can be remedied with data augmentation.

## 4.2 Evaluation on HPatches [3] – Fig. 5

HPatches contains 116 scenes with 6 images each. These scenes are strictly planar, containing only viewpoint or illumination changes (not both), and use homographies as ground truth. Despite

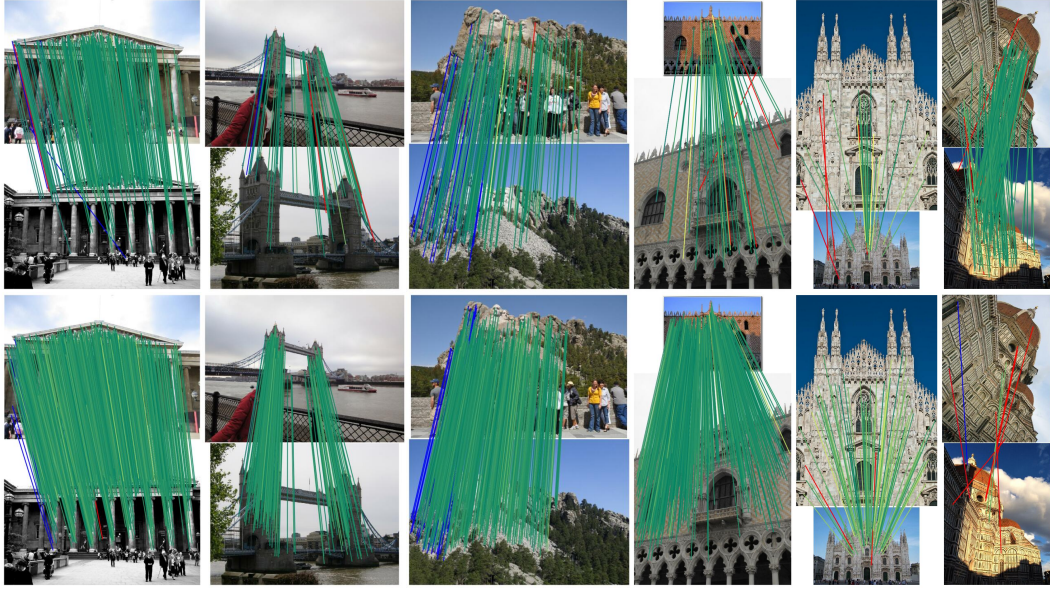

Figure 3: **Stereo results on the Image Matching Challenge (2k features).** Top: DoG w/ Upright HardNet descriptors [25]. Bottom: DISK. We extract cycle-consistent matches with optimal parameters and feed them to DEGENSAC [7]. We plot the resulting inliers, from green to yellow if they are correct (0 to 5 pixels in reprojection error), in red if they are incorrect (above 5), and in blue if ground truth depth is not available. Our approach can match many more points and produce more accurate poses. It can deal with large changes in scale (4th and 5th columns) but not in rotation (6th column), which is discussed further in section 4.1 and Fig. 6.

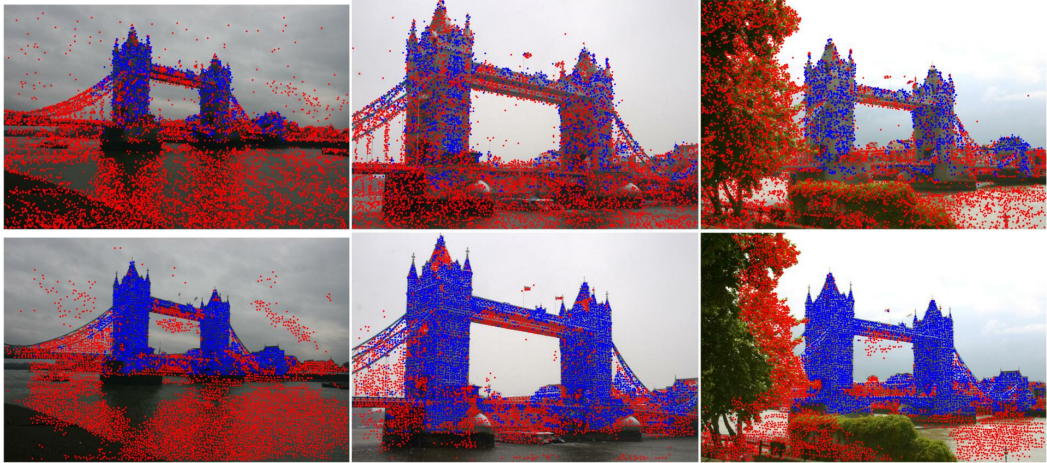

Figure 4: **Multiview results on the Image Matching Challenge (8k features).** Top: DoG w/ Upright HardNet descriptors [25]. Bottom: DISK. COLMAP is used to reconstruct the "London Bridge" scene with 25 images. We show three of them and draw their keypoints, in blue if they are registered by COLMAP, and red otherwise. Our method generates evenly distributed features, producing 76% more landmarks with 30% more observations per landmark than HardNet. Keypoints on water or trees have low scores and are rare among the top 2k features, but appear more often when taking 8k. This suggests that our method can reach near-optimal performance on a small budget.

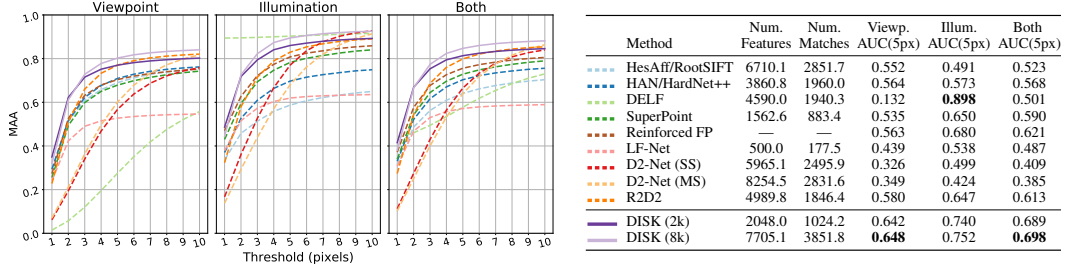

| Method | Num. Features | Num. Matches | Viewp. AUC(5px) | Illum. AUC(5px) | Both AUC(5px) |
|---|---|---|---|---|---|
| HesAff/RootSIFT | 6710.1 | 2851.7 | 0.552 | 0.491 | 0.523 |
| HAN/HardNet++ | 3860.8 | 1960.0 | 0.564 | 0.573 | 0.568 |
| DELF | 4590.0 | 1940.3 | 0.132 | **0.898** | 0.501 |
| SuperPoint | 1562.6 | 883.4 | 0.535 | 0.650 | 0.590 |
| Reinforced FP | — | — | 0.563 | 0.680 | 0.621 |
| LF-Net | 500.0 | 177.5 | 0.439 | 0.538 | 0.487 |
| D2-Net (SS) | 5965.1 | 2495.9 | 0.326 | 0.499 | 0.409 |
| D2-Net (MS) | 8254.5 | 2831.6 | 0.349 | 0.424 | 0.385 |
| R2D2 | 4989.8 | 1846.4 | 0.580 | 0.647 | 0.613 |
| DISK (2k) | 2048.0 | 1024.2 | 0.642 | 0.740 | 0.689 |
| DISK (8k) | 7705.1 | 3851.8 | **0.648** | 0.752 | **0.698** |

Figure 5: **Results on HPatches.** On the left, we report Mean Matching Accuracy (MMA) at 10 pixel thresholds. On the right, we summarize MMA by its AUC, up to 5 pixels. Results for RFP [6] were kindly provided by the authors, which explains why keypoint/match counts are missing.

its limitations, it is often used to evaluate low-level matching accuracy. We follow the evaluation methodology and source code from [11]. The first image on every scene is matched to the remaining five, omitting 8 scenes with high-resolution images. Cyclic-consistent matches are computed, and performance is measured in terms of the Mean Matching Accuracy (MMA), *i.e.*, the ratio of matches with a reprojection error below a threshold, from 1 to 10 pixels, and averaged across all image pairs.

We report MMA in Fig. 5, and summarize it by its Area under the Curve (AUC), up to 5 pixels. Baselines include RootSIFT [20, 2] on Hessian-Affine keypoints [24], a learned affine region detector (HAN) [26] paired with HardNet++ descriptors [25], DELF [28], SuperPoint [10], D2-Net [11], R2D2 [31], and Reinforced Feature Points (RFP) [6]. For D2-Net we include both single- (SS) and multi-scale (MS) models. We consider DISK with number of matches restricted to 2k and 8k, for a fair comparison with different methods.

We obtain state-of-the-art performance on this dataset, despite the fact that our models are trained on non-planar data without strong affine transformations. We use the same models and hyperparameters used in the previous section to obtain 2k and 8k features, without any tuning. Our method is #1 on the viewpoint scenes, followed by R2D2, and #2 on the illumination scenes, trailing DELF. Putting them together, it outperforms its closest competitor, RFP, by 12% relative.

### 4.3 Evaluation on the ETH-COLMAP benchmark [37] – Table 2

This benchmark compiles statistics for large-scale SfM. We select three of the smaller scenes and report results in Table 2. Baselines are taken from [6] and include Root-SIFT [20, 2], SuperPoint [10], and Reinforced Feature Points [6]. We obtain more landmarks than SIFT, with larger tracks and a comparable reprojection error. Note that this benchmark does not standardize the number of input features, so we extract DISK at full resolution and take the top ∼12k keypoints in order to remain comparable with SIFT. By comparison, a run on "Fountain" with no cap yields 67k landmarks.

### 4.4 Ablation studies and discussion

**Supervision without depth.** As outlined in Sec. 3, we use the strongest supervision signal available to us, which are depth maps. Unfortunately, this means we only reward matches on areas with reliable depth estimates, which may cause biases. We also experimented with a variant of $R$ that relies only on *epipolar constraints*, as in a recent paper [43]. We evaluate both variants on the validation set of the Image Matching Challenge and report the results in Table 3. Performance improves for multiview but decreases for stereo. Qualitatively, we observe that new keypoints appear on textureless areas outside object boundaries, probably due to the U-Net's large receptive field (see appendix). Nevertheless, this illustrates that DISK can be learned just as effectively with much weaker supervision.

**Non-maximum suppression and grid size.** The softmax-within-grid training time mechanism models the relative importance of features under a constrained budget, in a differentiable way. It can be replaced with an alternative solution, such as NMS, which we use at inference. In Table 4 we compare the training regime, where we sample at most one feature per grid cell, against the inference regime, where we apply NMS on the heatmap. We report results in terms of pose mAA on the validation set of the Image Matching Challenge in Table 4. For this experiment we removed the budget limit and took all features provided by the model. This shows that this inference strategy is

| Scene | Method | NL | TL | $\epsilon_r$ |
|---|---|---|---|---|
| Fountain | Root-SIFT | 15k | 4.70 | **0.41** |
| | SP | **31k** | 4.75 | 0.97 |
| | RFP | 9k | 4.86 | 0.87 |
| | DISK | 18k | **5.52** | 0.50 |
| Herzjesu | Root-SIFT | 8k | 4.22 | **0.46** |
| | SP | **21k** | 4.10 | 0.95 |
| | RFP | 7k | 4.32 | 0.82 |
| | DISK | 11k | **4.71** | 0.48 |
| South Building | Root-SIFT | 113k | 5.92 | **0.58** |
| | SP | **160k** | 7.83 | 0.92 |
| | RFP | 102k | 7.86 | 0.88 |
| | DISK | 115k | **9.91** | 0.59 |

Table 2: **Results on ETH-COLMAP [37]**. We compare Root-SIFT [20], SuperPoint [10], Reinforced Feature Points [6], and DISK. We report: **(NL)** number of landmarks, **(TL)** track length (average number of observations per landmark), and ($\epsilon_r$) reprojection error.

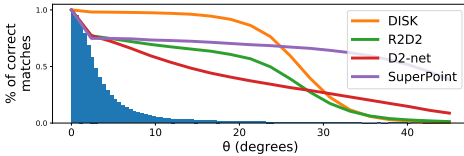

Figure 6: **Rotation invariance vs. rotations in data.** We report the ratio of correct matches between a reference images and their copies rotated by $\theta$. Overlaid is a histogram of relative image rotations in IMC2020-val.

| Variant | 2k features | | 8k features | |
|---|---|---|---|---|
| | Stereo | Multiview | Stereo | Multiview |
| Depth | **0.7218** | 0.8325 | **0.7767** | 0.8628 |
| Epipolar | 0.7145 | **0.8465** | 0.7718 | **0.8749** |

Table 3: **Ablation: match supervision.** We compare mAA on the Image Matching Challenge validation set, for DISK models learned with pixel-to-pixel supervision or epipolar constraints.

| Variant | Num. features | Num. matches | Stereo mAA($10^0$) | Multiview mAA($10^0$) |
|---|---|---|---|---|
| 1-per-cell | 5456.8 | 796.5 | 0.74774 | 0.84685 |
| NMS 3×3 | 8434.6 | 1699.9 | **0.77833** | 0.86864 |
| NMS 5×5 | 7656.0 | 1547.9 | 0.77657 | **0.87622** |
| NMS 7×7 | 6423.4 | 1271.1 | 0.77070 | 0.85642 |
| NMS 9×9 | 4946.2 | 942.0 | 0.75558 | 0.85362 |

Table 4: **Ablation: NMS.** We compare the feature selection strategy used for training (top) with NMS at inference time. Here we use *all detected features*, rather than subsample by score.

| Grid \ NMS | 3×3 | 5×5 | 7×7 | 9×9 |
|---|---|---|---|---|
| 8×8 | 0.7751 | **0.7824** | 0.7778 | 0.7586 |
| 12×12 | 0.7576 | **0.7580** | 0.7502 | 0.7431 |
| 16×16 | 0.7213 | **0.7214** | 0.7120 | 0.6999 |

Table 5: **Ablation: NMS vs grid size**. We show mAA vs. grid & NMS size on IMC2020-val, capping the number of features to 2k.

clearly beneficial, despite departing from the training pipeline. In Table 5 we show how mAA varies with grid size used for training. A smaller grid is beneficial in terms of performance but increases the number of extracted features, leading to larger distance matrices and higher computational expense.

**Feature duplication at grid edges.** Experimentally, we observe that 19.9% of features from grid selection (training) have a neighbour within 2 px, which likely corresponds to double detections. This has three potential downsides. (1) Compute/memory is increased, due to unnecessarily large matching matrices. (2) It rescales $\lambda_{kp}$ w.r.t. its intuitive meaning. Imagine that some detections are *strictly duplicated*: both forward and backward probabilities will "split in half", but the total probability of matching the two locations remains constant – this means that learning dynamics are not impacted, other than $\lambda_{kp}$ acting more strongly (on a larger number of detections). (3) In reality, detections are *close by*, instead of duplicated, which may make the algorithm less spatially precise: since duplication means a failure of the sparsity mechanism, we learn in a regime where imprecise correspondences are more common than at inference, favoring shift-invariance in the descriptors more than desired. The results DISK attains on HPatches, including at a 1-pixel error threshold, and the very low reprojection error on the ETH-COLMAP benchmark, suggest that these do not pose a significant problem for performance.

## 5 Conclusions and future work

We introduced a novel probabilistic approach to learn local features end to end with policy gradient. It can easily train from scratch, and yields many more matches than its competitors. We demonstrate state-of-the-art results in pose accuracy for stereo and 3D reconstruction, placing #1 in the 2k-keypoints category of the Image Matching Challenge using off-the-shelf matchers. In future work we intend to replace the match relaxation introduced in Sec. 3, with learned matchers such as [46, 33].

**Acknowledgement.** This research was partially funded by Google's *Visual Positioning System*.

## Broader impact

There already are many applications that rely on keypoints, and although our method has the potential to make them more effective, we do not expect new, specific issues arising from our research. As all technology, it can also be used unethically. In this instance, use in visually guided missiles or localizing photographs without user consent, further compromising privacy on the web, could be of concern. More generally, all automation of data processing brings disproportionately larger gains for established players with access to such data and resources, furthering the imbalance in global competitiveness, despite the nominal openness of the research.

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
