[Supplementary Material]

# APPENDIX — DISK: Learning local features with policy gradient

Supplementary material for NeurIPS submission 1194.

## Training data

In order to avoid image pairs which are not co-visible and ones which are too easy, we use a simple procedure. For every image $I$ we access the set of their 3D keypoints $\{L\}_I$, as provided in the COLMAP [36] metadata for the dataset, and for each pair $A, B$ we compute the ratio

$$r = \frac{|\{L\}_A \cap \{L\}_B|}{\min(|\{L\}_A|, |\{L\}_B|)}$$

which we use as a proxy for co-visibility, and pick all pairs $A, B$ for which $0.15 \leq r \leq 0.8$. In order to obtain the image triplets used during training, we randomly sample a "seed" image $A$ and then two more images $B, C$ among those with which $A$ was paired, based on the ratio criterion described above. We do not enforce that $B$ and $C$ are co-visible with respect to the criterion. We perform this sampling until we obtain roughly 10k triplets per scene.

We manually blacklist scenes which overlap with the test subset of the Image Matching Challenge: '0024' ("British Museum"), '0021' ("Lincoln Memorial Statue"), '0025' ("London Bridge"), '1589' ("Mount Rushmore"), '0019' ("Sagrada Familia"), '0008' ("Piazza San Marco"), '0032' ("Florence Cathedral"), and '0063' ("Milan Cathedral"). We also blacklist scenes which overlap with the validation subset of the Image Matching Challenge: '0015' ("St. Peter's Square") and '0022' ("Sacre Coeur"), which we use for the purposes of validation and hyperparameter selection, as in [16]. Finally, as per [12], we blacklist scenes with low quality depth maps: '0000', '0002', '0011', '0020', '0033', '0050', '0103', '0105', '0143', '0176', '0177', '0265', '0366', '0474', '0860', and '4541', as well as automatically remove scenes which produced less than 10k co-visible triplets.

In effect, we train on 135 scenes yielding $\approx$ 133k co-visible triplets. The dataset is available for download at at https://datasets.epfl.ch/disk-data/index.html.

## Continuous evaluation

With so many co-visible triplets a single iteration through the dataset (epoch) would take very long. In order to continuously evaluate performance of the model, we pause every 5k optimization steps (10k triplets) and evaluate stereo performance. To do so, we re-implement the mAA($10^o$) metric used by the benchmark [16], and apply it to a smaller subset of the validation set. We pick our best model according to this metric and then proceed with hyper-parameter tuning as described in Sec. 4.1. Our highest-performing model was obtained after 300k optimization steps.

## Computational cost

Our code, implemented in PyTorch [30], is run on an NVIDIA V100 GPU, with F32 precision. At inference time we obtain $\approx$ 7 frames per second for $1024 \times 1024$ input and training with $768 \times 768$ input requires $\approx$ 1.2 seconds per two triplets. Our code release includes an option to reduce the memory requirements through gradient accumulation, allowing for training with 12 Gb GPUs.

## Qualitative results for epipolar supervision – Fig. 7

As outlined in Sec. 4.4, our models may be supervised with pixel-to-pixel correspondences in the form of depth maps, or with simple epipolar constraints. With the latter, points appear around 3D object boundaries, as illustrated in Fig. 7. For simplicity, the main paper focuses on models trained with depth-based supervision.

## Breakdown by scene for the Image Matching Challenge [16] – Tables 6 and 7

We break down out results per scene in Table 6, for 2k features, and Table 7, for 8k features. Values copied from the challenge leaderboards (submissions #708 and #709).

(a) "Sacre Coeur" w/ depth-based supervision

(b) "Sacre Coeur" w/ epipolar-based supervision

(c) "Saint Peter's Square" w/ depth-based supervision

(d) "Saint Peter's Square" w/ epipolar-based supervision

Figure 7: **Qualitative results: depth vs epipolar supervision.** With depth-based supervision, our models learn to (usually) avoid textureless areas such as the sky. With epipolar-based supervision, points appear on the boundaries of 3D objects. They may or may not be matched: see for instance the obelisk on the rightmost images for (c) and (d). Thin structures, such as the lamp-posts on the leftmost images for (a) and (b), create features with epipolar supervision but not with depth supervision, presumably because they are typically absent in the depth maps. To illustrate this point we use the validation set from the Image Matching Challenge, following the same convention as in Fig. 4.

| | | Task 1: stereo | | | Task 2: Multiview | | |
|---|---|---|---|---|---|---|---|---|
| Scene | Num. Features | Input Matches | Num. Inliers | mAA(10°) | Input Matches | Num. Landmarks | Track Length | mAA(10°) |
| British Museum | 2048.0 | 717.9 | 571.9 | 0.4199 | 716.4 | 2223.5 | 6.55 | 0.6947 |
| Florence Cathedral | 2048.0 | 514.8 | 400.2 | 0.6922 | 530.8 | 2630.4 | 5.27 | 0.7563 |
| Lincoln Memorial Statue | 2048.0 | 430.9 | 326.5 | 0.5909 | 461.2 | 2098.4 | 5.66 | 0.8561 |
| London Bridge | 2048.0 | 452.3 | 350.8 | 0.5857 | 544.3 | 2009.8 | 6.19 | 0.8078 |
| Milan Cathedral | 2048.0 | 685.6 | 555.8 | 0.5267 | 672.1 | 2525.4 | 6.15 | 0.6840 |
| Mount Rushmore | 2048.0 | 471.7 | 392.6 | 0.3786 | 463.5 | 2534.8 | 4.88 | 0.5356 |
| Piazza San Marco | 2048.0 | 341.6 | 265.1 | 0.2603 | 338.9 | 2726.1 | 4.28 | 0.6033 |
| Sagrada Familia | 2048.0 | 474.2 | 366.6 | 0.5770 | 466.0 | 2696.9 | 5.10 | 0.8107 |
| St. Paul's Cathedral | 2048.0 | 539.1 | 408.3 | 0.5870 | 554.1 | 2407.0 | 5.83 | 0.7949 |
| Average | 2048.0 | 514.2 | 404.2 | 0.5132 | 527.5 | 2428.0 | 5.55 | 0.7271 |

Table 6: **Image Matching Challenge: Breakdown by scene (2k features).** We report results for each of the 9 scenes, and their average.

| | | Task 1: stereo | | | Task 2: Multiview | | |
|---|---|---|---|---|---|---|---|---|
| Scene | Num. Features | Input Matches | Num. Inliers | mAA(10°) | Input Matches | Num. Landmarks | Track Length | mAA(10°) |
| British Museum | 7839.8 | 1990.9 | 1530.4 | 0.4986 | 2002.2 | 6657.3 | 6.67 | 0.7377 |
| Florence Cathedral | 7996.9 | 1864.7 | 1415.9 | 0.7246 | 1927.3 | 8609.1 | 5.84 | 0.7840 |
| Lincoln Memorial Statue | 7597.3 | 948.2 | 649.4 | 0.6249 | 1029.8 | 5977.8 | 5.39 | 0.8851 |
| London Bridge | 7421.6 | 1073.3 | 811.7 | 0.6312 | 1333.9 | 5297.3 | 6.28 | 0.8208 |
| Milan Cathedral | 7887.5 | 2165.3 | 1703.2 | 0.5764 | 2135.0 | 7381.2 | 6.77 | 0.7031 |
| Mount Rushmore | 7976.1 | 1996.3 | 1612.9 | 0.4394 | 1961.5 | 8209.1 | 5.92 | 0.6103 |
| Piazza San Marco | 7999.0 | 1141.2 | 871.2 | 0.2842 | 1136.8 | 8675.5 | 4.53 | 0.5812 |
| Sagrada Familia | 7982.0 | 1870.3 | 1408.0 | 0.6170 | 1841.1 | 9154.7 | 5.80 | 0.8260 |
| St. Paul's Cathedral | 7897.3 | 1546.7 | 1144.0 | 0.6300 | 1606.6 | 7393.8 | 6.09 | 0.8039 |
| Average | 7844.2 | 1621.9 | 1238.5 | 0.5585 | 1663.8 | 7484.0 | 5.92 | 0.7502 |

Table 7: **Image Matching Challenge: Breakdown by scene (8k features).** We report results for each of the 9 scenes, and their average.