[Reviews · NeurIPS 2020]

Review 1

Summary and Contributions: This paper introduced a novel method “DISK” to match local features and aim to learn it through end-to-end reinforcement learning fashion. The paper is mainly to solve stereo and multi-view reconstructions and claimed to outperform all methods in stereo and multiview geometry on three different datasets as well as the ablation studies.

Strengths: This paper proposed several improvement in the design, but mostly are the incremental engineering approaches. For example, for the feature distribution, the paper based on a U-net and SuperPoint, retain the spatial structure and interpret cell in both a relative and an absolute manner. For the Match distribution, the paper proposed to relax cycle-consistent matching. The authors also defined some reward functions, imposed a penalty in gradient estimator, to name a few. In the perspective of significance, based on the Image Matching Challenge results, the proposed algorithm achieved good rankings in several benchmarks. For example, the paper obtained state-of-the-art performance on HPatches. The experiment and results are adequate as well, which is of interest for some of the NeurIPS community with interest in computer vision.

Weaknesses: For the mathematical aspect, this paper improved a lot of feature and matching techniques, but was lack of solid mathematical proofs. It is more of an engineering improvement towards this problem domain. For the experimental aspect, this paper introduced a lot of hyper-parameters without justification. For example, in line 178, how did the author choose the learning rate of ADAM = 1e-4? Did the author fine-tuned the hyper-parameters? The hyper-parameters problem is similar in line 172 for lambdas. As claimed also by the author, the proposed algorithm is not rotation sensitive (Figure 3), which is also an important weakness that need to be considered.

Correctness: Yes. The empirical methodology is correct.

Clarity: Overall the paper is clearly written.

Relation to Prior Work: The paper proposed DISK method based on previous feature extraction network method (U-net, SuperPoint), but further improved many details such as feature distribution, match distribution, reward function, and gradient estimators. Overall this paper made a moderate innovation than previous contributions, however, the limitation is lack of mathematical innovations.

Reproducibility: Yes

Additional Feedback:


Review 2

Summary and Contributions: This work proposes a new way to learn deep local features and descriptors. Α probabilistic feature sampling and matching framework is designed and trained with reinforcement learning. In contrast to recent RL-based approaches that use a single global reward, which is a weaker supervision signal, the proposed method uses a reward at local level, per feature match.

Strengths: - Using a reward at local level is a novel approach and is likely to be a key ingredient of the proposed approach. - The authors integrate in their training relaxed variants of standard matching approaches, such as the reciprocity of descriptor nearest neighbors. At inference, this is switched to the extract constrained and the relaxation is removed. This is one step to bringing the inference process closer to the training process. - The proposed method achieves good performance in recent and challenging benchmarks.

Weaknesses: - The paper mainly focuses on comparisons with prior work and does little work on ablations and revealing the key ingredients of the method and any underlined sensitivity. The achieved scores are good and good to know, but I m learning little from what moves us forward. For instance: -- How important is the probability of reciprocity and how much better is it compared to simply using a function of the descriptor similarity? -- what is the usefulness of the relative and absolute importance of features? Can they be used separately too, i.e. only one at a time? -- what is the impact of the grid size and does this correlate at all with the NMS window size? Meaning, will using less features for inference work better if the training has sparser sampling too? - Using a grid based selection must be prone to features locking on the boarders often. This is not handled by the way that authors pick one of the two features when they are on the borders and are overlapping. How often does this happen? Can the authors provide some statistics? Can the authors discuss whether this is a serious problem for the method or not? ---- the rebuttal has tried to address most of the concerns above. I think the paper deserves to be published and I remain with the same positive rating.

Correctness: I did not find any flaws

Clarity: The paper is in general well written. Probably a bit densely written when it comes to the method presentation, which is probably due to the limited space. Otherwise, it would be good to present concepts and components in a more detailed way.

Relation to Prior Work: The method discusses the relevant prior work and well positions the paper. The discussion included very recent work that is relevant and is also based on RL. The key differences are well discussed and motivated.

Reproducibility: No

Additional Feedback: questions to the authors are listed in weaknesses.


Review 3

Summary and Contributions: This paper contributes a novel deep learning based keypoint detector and feature extractor. The model is trained with RL, using a policy gradient method to maximize the expected matching reward. A key insight is that probabilistic relaxation of cycle-consistent matching gives exact gradients, eliminating the need for sampling and thus one source of gradient variance. The model can be trained from scratch, and performs competitively in the Image Matching Challenge, and on two other benchmarks.

Strengths: + The RL formulation based on probabilistic relaxation of cycle-consistent matching is novel. The model can learn keypoint detector and feature extractor jointly, and thanks to the reduced gradient variance, can be trained from scratch - which seems to be a significant result in itself. + Empirically, very competitive results on the Image Matching Challenge, and state-of-the-art on the other two benchmarks. + Compared to other works, the proposed model can generate significantly more inliers/landmarks, therefore it has the potential to further improve task performance when incorporating a learned matcher.

Weaknesses: - The paper should have more discussion on failure cases and limitations of the work. For example, Figure 3 indicates a lack of robustness against rotation, which the paper claims "could be solved with more data augmentation" - but I'd like to know more. Is rotation augmentation used in training? What are the considerations? Does it (or would it) help or hurt final performance?

Correctness: Yes

Clarity: Yes. This paper is a joy to read.

Relation to Prior Work: Yes

Reproducibility: Yes

Additional Feedback:


Review 4

Summary and Contributions: A local feature learning scheme is proposed in this paper. Leveraging reinforcement learning, the discrete keypoints are learned by an end-to-end deep neural network. Evaluated on the 2020 Image Matching Challenge and multiple benchmark datasets, the proposed method shows its effectiveness.

Strengths: Target on learning a discrete task, the proposed method explores learning local image features using policy gradient. Instead of estimating every correct match, the objective is relaxed as finding correct feature matches. Towards the objective, the feature and match distributions are configured. A relax cycle-consistent matching is utilized to refrain from ambiguous matching in training. The overall pipeline is reasonable and well described in the paper. The experimental part shows strong results against other methods. Sufficient comparison experiments and ablation study are provided. The proposed method also demonstrates itself in the 2020 Image Matching Challenge.

Weaknesses: My main concerns about this paper are as follow: (1) The main contribution of the paper is not well discussed. It is not the first work using reinforcement learning to learn local features. But what is the most important difference between the proposed method and the others? Using policy gradient for discrete task learning is a reasonable idea. However, more theoretical supports are necessary. The method section is more like a processing introduction that is lacking of motivation. (2) Analysis about the components in the learning and inference stages are not sufficient. There are many details in the proposed system, for example, the relaxation of the learning objective, using the cycle-consistent matching for building the match distribution and settings in the gradient estimator. How do these components influence the final result? Which part obtains the maximum performance gain? (3) The experimental part shows the strong effectiveness of the learned features. However, the result illustration is mostly a numerical exhibition. Some of the experimental phenomenon are not well explained. For example, the learned features perform well in scale changing cases but are sensitive to object rotation. The authors explain that these can be improved by doing data augmentation. But a simple introduction of the dataset distribution may be more convincing. And in the discussion of the “Supervision without Depth”, key points on textureless areas are hypothetically explained as the cause of the U-Net’s receptive field. This is not a satisfied hypothesis, and it is not well supported in the appendix.

Correctness: Yes, the claims and method are in general correct.

Clarity: The paper is well organized. And the presentation is clear.

Relation to Prior Work: The proposed method is mostly related to references [7] and [9]. But the authors should provide more analysis for their difference from the previous contributions.

Reproducibility: Yes

Additional Feedback: I suggest the authors provide more analysis for their difference from previous contributions and more discussions about the motivation of using policy gradient. My major concern about the motivation and difference from the prior arts are addressed in rebuttal. So I change my overall score to 6.

[Author Response · NeurIPS 2020]

1 **[Submission 1194: "DISK"]** We thank all reviewers for their insightful comments, and address their concerns.

Figure A: Rotation invariance vs. rotations in data.

| NMS Cell | $3\times3$ | $5\times5$ | $7\times7$ | $9\times9$ |
|---|---|---|---|---|
| $8\times8$ | 0.7751 | **0.7824** | 0.7778 | 0.7586 |
| $12\times12$ | 0.7576 | **0.7580** | 0.7502 | 0.7431 |
| $16\times16$ | 0.7213 | **0.7214** | 0.7120 | 0.6999 |

Figure B: mAA vs. cell size & NMS on IMW2020 (val).

**R1, R5: domain-specific engineering & lack of mathematical innovations.** Only one other work applies policy gradient to local features [7]. It relies on non-differentiable methods based on assumptions on the pre-trained model ([7], Sec. 3.3, points 1 and 2). Instead, we optimize a simpler objective function and exploit its structure to reduce gradient variance (L129-133 in our paper), allowing us to train from scratch, unlike [7] that we outpeform on two datasets. In short, ours is the first learned, end-to-end method to outperform well-tuned baselines using hand-crafted detectors. Finally, please note that [7] was officially published after the NeurIPS submission deadline, which by NeurIPS guidelines makes it a contemporaneous submission.

**R1: DISK is based on previous work (U-Net, SuperPoint) and only offers moderate innovation.** The only similarity with SuperPoint is that we also use a CNN to densely find keypoint score maps and descriptors. SuperPoint is not a RL method and uses neither feature/match *distributions* nor a *reward* function. We used a U-Net because it is a proven architecture and our focus was more on the RL algorithm than on developing a specific architecture.

**R1, R4, R5: Rotation invariance.** Limited rotation invariance is a *deliberate* choice, because rotation estimation is counterproductive for upright images: see [14] (Sec. 6.5, Tables 10-11). As an experiment, we randomly pick 36 images from the IMW2020 test set, and extract and match features between them and their copies, rotated by $\theta$. We compute the ratio of correct matches (within a 3px threshold) and show it in Fig. A. We also superimpose a histogram of relative image rotations between all pairs of images on the IMW2020 validation set. Our current approach is *extremely* robust to the rotations found in the data, which could be further increased by data augmentation. We will clarify this in the paper.

**R1: Hyperparameters.** We have relatively few of them: (1) cycle-consistency temperature $\theta_M$ (2) true positive reward $\lambda_{tp}$ (3) false positive penalty $\lambda_{fp}$ (4) detection cost $\lambda_{kp}$. Aside from the initial annealing of $\lambda_{fp}$ and $\lambda_{kp}$ (L172-175), we did not find the method sensitive to hyperparameters, including ADAM LR. They were chosen arbitrarily and found to work well. We tuned inference parameters (NMS window & RANSAC settings) by search, as described in L194-197.

**R1, R3, R5: What is the contribution of individual components of the pipeline? Can they be replaced?** We do not view DISK as a series of independent components. Because we maintain a probabilistic interpretation throughout the pipeline, we can easily reason about the effect of hyperparameters $\lambda_{tp}, \lambda_{fp}$ and $\lambda_{kp}$. An ablation study, such as replacing our matching scheme with a margin loss [23], would require "plumbing" to balance the respective loss terms, making the comparison unreliable. We experimented with an alternative matching relaxation, using the entropy of the match distribution as a proxy for confidence (in place of cycle-consistency). It performed comparably while requiring more hyperparameters and computation, and we dropped it from the submission due to space constraints and simplicity.

**R3: Relative vs. absolute importance of features.** Absolute importance measures keypoint quality. Relative importance is a mechanism to enforce feature sparsity in a differentiable manner. Absolute importance can be paired with a different sparsity mechanism – in fact, for inference we replace relative importance with NMS.

**R3: Cell size vs. NMS.** We find models trained with $8 \times 8$ to outperform larger grid cells, regardless of NMS window. Fig. B summarizes this for different settings, on IMW2020. For brevity, we average stereo and multiview performance.

**R3: Feature "duplication" on cell borders.** Experimentally, we observe that 19.9% of features from grid selection (training) have a neighbour within 2 px. This has three potential downsides. (1) Compute/memory is increased, due to redundancies. (2) It rescales $\lambda_{kp}$. Imagine that some detections are *strictly duplicated*. The probability of matching two locations remains constant – this means that learning dynamics are not impacted, other than $\lambda_{kp}$ acting more strongly (on a larger number of detections). (3) Detections are *close by*, instead of duplicated, which may make the algorithm less spatially precise: since duplication means a failure of the sparsity mechanism, we learn in a regime where imprecise correspondences are more common than at inference, slightly favoring shift-invariance in the descriptors. However, DISK is #1 on HPatches, even at a 1px threshold, and attains very low reprojection error on ETH-COLMAP benchmark.

**R5: Features on textureless areas.** We claim that features *outside object boundaries* are matched using contextual information. Fig. 6 of the appendix illustrates this with detections on the sky (many of them matched – blue dots) near objects of interest. Since the sky has no intrinsic features, only the spatial context could be used to match them.

**R5: Motivation for policy gradient and relation to [7] and [9].** Please note we discuss this in L18-L30 and L51-65.

[Meta-Review · NeurIPS 2020]

The paper presents a new technique for learning feature matching using reinforcement learning. Strengths are good experimental results showing the technique is effective in several problems. Weaknesses are some concerns about positioning (motivation, related work), but these are addressed in the rebuttal. The final reviews are uniformly positive. This is a clear accept.